# Microstructure and Room Temperature Mechanical Properties of Different 3 and 4 Element Medium Entropy Alloys from HfNbTaTiZr System

**DOI:** 10.3390/e21020114

**Published:** 2019-01-26

**Authors:** Jiří Zýka, Jaroslav Málek, Jaroslav Veselý, František Lukáč, Jakub Čížek, Jan Kuriplach, Oksana Melikhova

**Affiliations:** 1UJP PRAHA a.s., Nad Kamínkou 1345, 156 10 Praha 5—Zbraslav, Czech Republic; 2Department of Materials Engineering, Institute of Plasma Physics CAS, Za Slovankou 1782, 1820 Praha 8, Czech Republic; 3Faculty of Mathematics and Physics, Charles University, V Holešovičkách 2, 18000 Praha 8, Czech Republic

**Keywords:** refractory high entropy alloys, medium entropy alloys, mechanical properties, microstructure

## Abstract

Refractory high entropy alloys (HEA) are promising materials for high temperature applications. This work presents investigations of the room temperature tensile mechanical properties of selected 3 and 4 elements medium entropy alloys (MEA) derived from the HfNbTaTiZr system. Tensile testing was combined with fractographic and microstructure analysis, using scanning electron microscope (SEM), wavelength dispersive spectroscope (WDS) and X-Ray powder diffraction (XRD). The 5 element HEA alloy HfNbTaTiZr exhibits the best combination of strength and elongation while 4 and 3 element MEAs have lower strength. Some of them are ductile, some of them brittle, depending on microstructure. Simultaneous presence of Ta and Zr in the alloy resulted in a significant reduction of ductility caused by reduction of the BCC phase content. Precipitation of Ta rich particles on grain boundaries reduces further the maximum elongation to failure down to zero values.

## 1. Introduction

High entropy alloys (HEAs) attract attention of a growing number of scientists and researchers. A concept of mixing of 5 or more elements in equimolar or near equimolar concentrations is used in order to explore central regions of multicomponent alloy phase space [1]. This approach is driven by possibility to get stable solid solution microstructure of the alloy with favourable mechanical properties (high strength and good ductility) as well as other physical properties, for example, good corrosion resistance [2]. Microstructure stability shall be provided by high configurational entropy of the system, supposing random arrangement of alloying elements [3,4]. Although increased configurational entropy may in principle stabilize solid solution its stabilizing effect is usually found insufficient to counteract driving forces for formation of intermetallic phases [5,6].

Since the pioneering work of Yeh [4], original concept has been evolved and widened. The term HEA is connected mainly with approach utilizing high configurational entropy to get single phase solid solution of multiple elements. Multiple principal element alloys (MPEA) represent a broader approach when the main motivation is exploring of a vast composition space of multi-principal element alloys without primary concern about the magnitude of the configurational entropy [1]. MPEAs with multi-phase microstructures are denoted complex concentrated alloys (CCA) [1].

Different elements are used to fabricate HEAs. Although the number of possible 4 or 5 elements combinations is enormous five basic metallic HEA groups can be distinguished: FCC HEAs based on the Cantor alloy, BCC refractory metal HEAs, light element HEAs, HCP HEAs and precious functional HEAs [7]. Ceramic HEAs have been established as well [8,9].

Our attention was attracted by the refractory HEA group, which is composed of elements from IV, V and VI groups of the periodic table of elements [10]. These elements are characterized by prevailing good mutual miscibility and high melting points. The melting point of titanium (1668 °C) is the lowest among them. Therefore HEAs composed of these elements are intended for high temperature applications. Moreover some of these elements are biocompatible [11]. Therefore some of these alloys can be attractive for bioimplant related materials research. Note that β-Ti alloys used in bioimplant research are usually composed of this group of elements [12]. Several biocompatible HEAs have been already studied [13,14,15,16]. Also, hard ceramic composites were produced from the metals from IV, V and VI groups of the periodic table [17].

Original definition of HEAs with single phase solid solution microstructure and good room temperature mechanical properties, especially ductility, is beneficial also for use in human medicine. Inspiration can be found also in other materials used for production of bioimplants which are used or have been used in other parts of material research, for example, TiAl6V4 in aerospace, CoCrMo in aero engines.

Our first attempt in research of HEAs was therefore related to HfNbTaTiZr refractory metal alloy [18]. Our research was inspired by Senkov [19] and confirmed our expectations. Ingots produced by vacuum arc melting possessed single phase solid solution microstructure, with a high room temperature tensile strength and ductility. 

Some of the elements used in the HfNbTaTiZr alloy are very similar to each other with very similar chemical behaviour; Nb is chemically similar to Ta and Zr to Hf. Therefore a question has emerged whether it is necessary to use all of these elements in order to get random solid solution and good mechanical properties (high strength and sufficient ductility). 

The design of the present experiment was as follows. First, we reduced the original HfNbTaTiZr alloy system by removing Hf, which is almost identical to Zr from the chemical point of view, then we distracted Ta, which is very similar to Nb; thus getting NbTiZr alloy with equiatomic concentrations. Then we produced also other 3 element combinations containing Ti, namely TaTiZr and NbTaTi. Based on the observed results; 4 element alloys containing Ti and Ta with different Nb to Zr ratio were produced as well to explore the influence of chemical composition on elongation. 

NbTaTiZr [14] and NbTiZr [20,21] studies have already been reported. No reports about TaTiZr and NbTaTi were found in literature.

One can notice that set of the produced element combinations is not complete. The motivation for element selection was as follows. Titanium was used in all combination because it is element with the lowest density. Hafnium was not used since it is very reactive in ambient atmosphere and it is hard to get Hf free of impurities. Both hafnium and tantalum are high density and high price element; thus less suitable for practical use.

## 2. Materials and Methods 

Experimental alloys were prepared by vacuum arc melting in water cooled copper crucible. Chemical purity of inserted elements was 99.9%. Final cast ingot has approximately 100 mm in length, 30 mm in width, 10 mm in height and 400 g in weight. Casting was performed 8x times and flipped for each melt to mix the elements thoroughly and suppress chemical heterogeneity. All investigated alloys were prepared at the same equipment under the same conditions. Nor HIP, nor other heat treatment was applied to the cast material. 

Tensile test bodies 5 mm in diameter and 25 mm of measured length were strained with the strain rate of 2 × 10^−4^ s^−1^ using Instron 1185 testing machine equipped with video extensometer. 

Metallographic sections perpendicular to the length of the casting were prepared. Vickers hardness HV30 was measured using Zwick ZHU 250 Topline universal testing machine. 

Fractography analysis of broken specimen surfaces was carried out on a scanning electron microscope (JEOL JSM 7600F).

Microstructure was examined by scanning electron microscope (FEI Quanta 200F) equipped with wavelength dispersive X-ray spectroscope (WDS) for local chemical composition analysis.

In order to determine phase composition and structures of phases in the alloys, powder X-ray diffraction (PXRD) analysis was performed. A Bruker D8 Discover diffractometer, CuKα radiation and 1D detector were used. Lattice parameters and phase composition were determined by Rietveld analysis of PXRD diffraction patterns using TOPAS V5 code [22]. 

*Ab-initio* quantum Monte Carlo (MC) simulation was performed to obtain atomic configuration of the 5 element HfNbTaTiZr alloy. Hf, Nb, Ta, Ti, Zr ions were distributed randomly in equimolar concentration into a 250 atom BCC supercell. This initial state was relaxed with respect to ion positions until minimum of the total free energy was reached. The equilibrium configuration corresponding to minimum free energy was obtained using a Metropolis MC algorithm at 300 K (room temperature). The details of the simulation and more complete results will be published elsewhere.

## 3. Results

### 3.1. Chemical Composition

Chemical compositions of investigated alloys in atomic percentages are given in Table 1. Important HEA related parameters are listed in Table 2. Valence electron concentration—VEC, mixing enthalpy−∆H_mix_, difference in atomic radii−δ, Ω parameter, mixing entropy−S_mix_ are calculated in accordance with literature [1], where T_m_ is a calculated average melting point. According to entropy-based definition HEAs are characterized by the configurational entropy higher than 1.61 *R* [1]. Hence, according to this definition, only HfNbTaTiZr alloy is high entropy alloy. The other alloys studied in the present work are medium entropy alloys.

### 3.2. Mechanical Properties

Tensile curves of investigated alloys performed at room temperature are shown in Figure 1. Results of tensile tests performed and hardness measurements performed at room temperature are given in Table 3.

The 5 element HEA HfNbTaTiZr alloy has the best tensile properties among the alloys studied, both regarding the strength and elongation. The 4 element NbTaTiZr alloy has similar strength and hardness as HfNbTaTiZr but elongation is reduced to 6.4%. The NbTaTi has low strength, half of that for HfNbTaTiZr but the highest elongation. The NbTiZr alloy has strength values below 1000 MPa but elongation is relatively high (14.2%). The TaTiZr alloy exhibited brittle behaviour, breaking before reaching the yield point. The Nb_0.5_TaTiZr_1.5_ alloy performed very similarly, breaking also before reaching yield point but a little bit later than TaTiZr alloy. The Nb_1.5_TaTiZr_0.5_ alloy reached the yield point (822 MPa) but the elongation was 0.33% only.

Highest hardness (HV30 = 485) values were measured in the most brittle alloys TaTiZr and Nb_0.5_TaTiZr_1.5_. Hardness of the 4 and 5 element equimolar alloy was very similar to each other, lower by ≈130 HV compared to the brittle alloys. Hardness of the NbTiZr and Nb_1.5_TaTiZr_0.5_ are slightly lower than 300 HV. The most ductile alloy NbTaTi exhibited the lowest hardness of 246 HV.

### 3.3. Fractographic Analysis

Fracture surfaces of specimen broken during tensile test were subjected to fractographic analysis, see Figure 2. Both transgranular and intergranular ductile fracture can be found on fracture surfaces of the NbTiTaZr alloy, see Figure 2a–c. Elongated particles were observed on the intergranular ductile fracture surface. Ductile dimples of different diameters were found on the transgranular fracture surface of NbTaTi alloy, see Figure 2d. Transgranular ductile fracture was found also in the case of NbTiZr alloy (Figure 2f). On the other hand, fracture surface of TaTiZr alloy is dominantly flat and brittle and containing number of needle like particles, see Figure 2e. River-like pattern is present as well (not shown). Nb_1.5_TaTiZr_0.5_ specimen fracture surfaces are composed of transgranular ductile fracture and intergranular fracture, see Figure 2g. A lot of particles are present on the intergranular part of the fracture surface. Nb_0.5_TaTiZr_1.5_ fracture surfaces (Figure 2h–i) are similar to those of Nb_1.5_TaTiZr_0.5_ and NbTaTiZr alloy but without presence of particles on the intergranular part of the fracture surface.

### 3.4. Microstructure

Microstructure of the investigated alloys is shown in Figure 3. Grains size was estimated by light microscopy on metallographic specimens with mirror-like polished and slightly etched surface using the linear intercept procedure [23]. It revealed grain size around 0.5 mm which is similar for all alloys. All alloys, except of NbTiZr alloy, exhibited dendritic segregation.

Small submicron precipitates were found on grain boundaries of TaTiZr, Nb_1.5_TaTiZr_0.5_ and Nb_0.5_TaTiZr_1.5_ alloys, see Figure 4. WDS analysis of these precipitates revealed that they are rich in Ta. For example the average Ta concentration in the precipitates in TaTiZr alloy is (52 ± 1) at.% while the Ta content in the matrix of this alloy was found to be (38 ± 1) at.%.

Figure 5 illustrates existence of two phases. One of them appears brighter because of higher average Z number while the second one appears darker due to lower average Z number. Phase separation into these two phases occurs on the length scale of ~20 µm. The only exception is NbTiZr alloy, which shows no phase segregation, see Figure 5d. WDS line analysis showed enhanced concentration of Zr and Ti in the dark phase and Ta and Nb in the bright phase, see Table 4.

#### 3.4.1. XRD Analysis

XRD patterns in Figure 6 exhibit peaks in positions corresponding to reflection of a BCC phase but with different extent of broadening. HfNbTaTiZr [18], NbTaTi and NbTiZr alloys consist of single BCC phase. Rietveld refinement of XRD patterns suggested existence of two BCC phases with slightly different lattice parameters in case of NbTaTiZr and TaTiZr alloys (Table 5). This corresponds to the previously observed presence of the dendritic microstructure in the feedstock powder particles [24]. The regions that solidify earlier were enriched in Nb, Ta, thereby triggering a measurable change in the respective lattice parameters of the BCC regions as compared to the interdendritic regions with increased Zr, Ti content. Rietveld refinement suggested even three BCC phases in case of Nb_1.5_TaTiZr_0.5_, Nb_0.5_TaTiZr_1.5_ with similar lattice parameters (Table 5). Grain size cannot be calculated from XRD data since the broadening of XRD reflections caused by finite grain size was found to be negligible. It means the average grain size of the alloys studied is higher than 100 nm which is consistent with metallographic and SEM observations. Dominant source of peak broadening is chemical heterogeneity, that is, local changes of lattice parameter due to spatial variations of chemical composition.

#### 3.4.2. Monte Carlo Simulation of Microstructure 

*Ab-initio* MC simulation was performed to evaluate the microstructure stability of investigated alloys. Simulation was performed for the 5 element HfNbTaTiZr alloy, because it contains all elements considered in investigated alloys.

Hf, Nb, Ta, Ti, Zr ions were distributed randomly in equimolar concentration into a 250 atom BCC supercell. This initial state was relaxed with respect to ion positions until minimum of the total free energy was reached. The equilibrium configuration corresponding to minimum free energy was obtained using a Metropolis MC algorithm at 300 K (room temperature). The details of the simulation and more complete results will be published elsewhere. Figure 7 shows the equilibrium atomic configuration corresponding to the minimum of total energy. The most apparent effect is a rather one-dimensional Ta object (‘wire’) along the <100> direction. The Ta wire is surrounded predominantly by Nb ions. The rest of the simulation box is filled up by the mixture of Ti, Zr and Hf. The latter two elements appear to be well separated from the Ta and Nb region. Hence, this preliminary result indicates inhomogeneity of the HfNbTaTiZr alloy. Such inhomogeneities–though they need to be yet verified experimentally–could affect physical properties of the alloy.

## 4. Discussion

Mechanical properties tests have revealed that combination of Ta and Zr reduces elongation. Microstructure analysis revealed existence of dark and bright phases because of Ta-Zr segregation. This kind of dendritic segregation was reported elsewhere [25,26,27]. It was shown [25] that Ta with Nb segregates during solidification to the solid and Zr with Ti to the liquid.

Figure 8a shows relation between the elongation to failure, A and the total atomic fraction of Ta and Zr. All alloys studied exhibit linear relationship of A on the Ta + Zr concentration. The only exception is Nb_1.5_TaTiZr_0.5_ alloy not following the linear relationship because, it is brittle despite of relatively low Ta + Zr content. In the latter alloy Nb probably plays similar role as tantalum. Figure 8b shows the content of the BCC1 phase as a function of the total concentration of is Ta and Zr. Obviously it obeys similar linear relationship with the net concentration of Ta and Zr as the elongation. 

Intergranular nanosized precipitates were found on grain boundaries of brittle alloys with zero or almost zero elongation, namely TaTiZr, Nb_1.5_TaTiZr_0.5_ and Nb_0.5_TaTiZr_1.5_. XRD analysis revealed 2 BCC phases (BCC1, BCC2) which are probably caused by the microsegregation [24]. In two cases small amount of third BCC phase (BCC3) was detected, it can be connected with the intergranular precipitation. However, it was not investigated in detail in the present study.

Usually relations between the misfit parameter δ [1] and strength or hardness are reported, since rising δ shall indicate higher solid solution strengthening. Since in the present case we have single phase solid solution only in 3 alloys out of 7 investigated, the relation between hardness and δ is more complicated than simple linear dependence. Indeed if we exclude two most brittle alloys, indicated by red symbols in Figure 9a, the dependence of hardness and δ becomes rather close to the linear relationship. In elongation and δ relationship NbTaTi alloy, the most ductile one, destroys possible correlation as well, Figure 9b.

A relation between the elongation and the VEC parameter was reported in case of refractory HEAs [28]. Alloys with VEC lower than 4.4 shall be ductile, alloys with VEC higher than 4.6 shall be brittle. Figure 10a shows relation between elongation of investigated alloys and their VEC parameters. VEC values are close to the boundary value of 4.5. Brittle alloys TaTiZr (VEC = 4.329), Nb_0.5_TaTiZr_1.5_ (VEC = 4.375) and ductile alloy NbTaTi (VEC = 4.663) do not follow the reported rule. However, these alloys are MEAs and not HEAs. Brittle alloys with VEC lower than 4.5 are not single phase systems but contain 2 or 3 BCC phases and intergranular precipitates. NbTaTi is ductile because of absence of Ta-Zr combination in the alloy. The NbTiZr alloy exhibits similar behaviour. The HfNbTaTiZr alloy is also ductile, despite the fact that it contains Ta-Zr combination. Although MC simulation showed segregation of Ta, any mark of such segregation, was detected in experiment. When plotting the total elongation A as a function of the BCC1 phase content one can observe a clear linear relationship, see Figure 10b. Hence, there is a positive correlation between the total elongation and the BCC1 phase content. This is not surprising since both the total elongation and the BCC1 phase content decrease with Ta + Zr concentration, see Figure 8. The only exception is Nb_1.5_TaTiZr_0.5_ alloy which does not follow this trend.

The presence of other BCC phases influences also fracture mechanisms and fracture surfaces of tensile specimens.

It is important to investigate the question of existence of 2 or more BCC phases in our alloys. In HEA related research a lot of work has been done to establish connection between chemical composition or related parameters and microstructure, especially existence of single phase or intermetallics or amorphous microstructure. Satisfying δ < 0.066 and ΔH_mix_ > − 11.6 kJ/mol, however, is necessary but not sufficient conditions to form solid solutions in HEAs. Checking the binary phase diagrams among constituent elements can give some further guidance in designing solid solutions forming HEAs [28]. All our alloys satisfy these two conditions but some of them are not single phase. Thus binary diagram analysis is needed. 

Mixing enthalpies of element pairs relevant to the investigated alloys is shown in Table 6. Some mixing enthalpies are zero (Ti-Zr, Ti-Hf, Zr-Hf and Ta-Nb – pairs of elements from the same group) or almost zero (Ti-Ta, Ti-Nb), which is ideal for solid solution forming. Other mixing enthalpies are little bit higher (Ta-Zr, Ta-Hf, Nb-Zr, Nb-Hf – pairs from different but neighbouring groups). Above zero mixing enthalpies can cause existence of miscibility gaps in respective binary diagrams. Binary equilibrium phase diagrams of pairs with higher mixing enthalpies are shown in Figure 11. No intermetallic phases are present but large miscibility gaps can be found. Although in case of Ta-Zr diagram miscibility gap is in the range between 800 °C and 1780 °C, see Figure 10a, we found two BCC phases at room temperature.

It is not clear whether presence of BCC phases in our less ductile alloys is due to stabilizing by rapid cooling during vacuum arc melting or due to differences between binary alloy and 3, 4 or 5 element alloys. 

Existence of two BCC phases was reported in similar RHEA system with Mo and V instead of Ta and Nb [34]. CALPHAD simulation of the HfNbTaTiZr was performed, too [35]. A mixture of BCC and HCP phases with Ta Zr segregation was predicted. Experimental observation was performed on cold deformed and heat treated alloy in combination with high pressure torsion (HPT) and isothermal annealing in the range 300 °C–1100 °C and BCC + HCP phases were identified. Combination of two BCC phases and HCP was detected in a HfNbTaTiZr specimen annealed at 500 °C for 100 h. On contrary we observed only a single BCC phase in as-cast state.

System HfNbTaTi, similar to our NbTaTiZr alloy was calculated by CAPLHAD method [36]. Also combination of BCC + HCP phases was found to be stable at room temperature the BCC2 phase appeared between 750 °C and 1000 °C.

In Ref. [25] equimolar NbTaTiZr was modified by adding Ti and removing Ta and results similar to the present work were observed. The elongation to failure increased with rising titanium content. Dendritic segregation came from solidifying, segregation of Ta and Zr and intergranular precipitates were formed after 1200 °C/8 h annealing. But only single BCC phase was found in the as-cast state. Stabilizing of BCC phase by reduction of Ta content and rising Ti content was proposed [25]. A clear correlation between strength or elongation and the misfit parameter δ were established. It is probably due to single phase microstructure. 

Table 7 shows comparison of tensile properties of the alloys studied in the present work with MEAs reported in literature. There is a good agreement in ultimate tensile strength of NbTaTiZr alloy but referred rupture strength is higher and elongation much lower than in case alloy studied in this work.

Elongation, A calc., based on tantalum and zirconium combined content using equation from Figure 8a was calculated. There is a good correlation in higher Ti contents in case of alloys referred. But there is important difference in lower Ti content, although single phase solid solution for all compositions was reported [25].

This difference corresponds with important change in elongation with Ti fraction. No clear explanation of this phenomenon was given [25]. Comparing experimental details of the present study with Ref. [25] one can recognize that dimensions of ingots were different. Ingots of alloys studied in the present work were larger and thereby solidifies and cools slower than ingots in Ref. [25]. Therefore NbTaTiZr alloy containing 100% of the BCC1 phase reported in Ref. [25] was likely due to higher cooling rate; while in the present work the same alloy exhibits a mixture of BCC1 and BCC2 phases, see Table 4. On the contrary in Ref. [25] the NbTaTiZr is referred as brittle, we measured 6.4% elongation, see Table 3, however, fracture mechanism has changed from transgranular with ductile dimples to intergranular, see Figure 2. Different interstitials element content may cause the difference in deformation of nominally same alloy.

In this study we use the same furnace and the same size of ingots in case of all investigated alloys. Therefore it is solely the effect of Ta + Zr content what affects the stability of the BCC1 phase. The cooling rate was high enough to produce 100% BCC1 microstructure in case of 5 element HfNbTaTiZr alloy but not in the 4 element alloy NbTaTiZr.

Intensive precipitation at grain boundaries was observed in the brittle TaTiZr alloy, see Figure 4. This alloy contains highest combined content of Ta + Zr and a low amount of the BCC1 phase.

Similar microstructures were found in Refs. [37,38]. These works investigated the influence of middle temperature annealing on the 5 element HEA HfNbTaTiZr in deformed and homogenisation annealed state. Combination of BCC and HCP phases were detected, at specific conditions (longer annealing temperatures and longer times) Ta rich BCC and Zr rich HCP phase precipitates were formed. Similarity of the microstructures and phases suggests that these effects have similar origin, namely Ta+Zr content, partial decomposition of the BCC1 phase to the BCC2 because of relatively high Ta-Zr mixing enthalpy and resulting miscibility gap.

The 5 element HfNbTaTiZr alloy was studied in Refs. [35,36]. It has lower Ta + Zr content than NbTaTiZr, Nb_0.5_TaTiZr_1.5_ and TaTiZr studied in the present work. Ingots used in Refs. [35,36] were smaller compared to that prepared in the present work, thus gaining higher cooling rate. The ingots were also annealed for homogenization and water quenched, thus metastable BCC1 solid solution was obtained. However, subsequent mid temperatures annealing caused decomposition of the BCC1 solid solution.

In our study, alloys are destabilized by higher Ta + Zr content and the cooling rate was slower than in other studies, therefore the BCC1 phase decomposition took place already during cooling. The MC simulation revealed segregation of Ta in the equilibrium state of the HfNbTaTiZr alloy. This supports the idea of Ta + Zr destabilizing HEAs also during ageing annealing. Thus single phase BCC1 structure is metastable in the 5 element HfNbTaTiZr HEA alloy at room temperature and can be preserved after high temperature annealing by rapid cooling. The role of rapid cooling has been discussed in the review [1].

The alloys containing the BCC2 phase are free of Hf. An HCP phase was predicted and confirmed experimentally in Hf containing alloys [35,36,37,38]. It means that Hf acts as an element stabilizing HCP phase.

Further research shall be focused on detail investigations of the BCC2 and BCC3 phases and intergranular precipitates detected in some alloys. Destabilising effect of Ta + Zr on the BCC phase stability during manufacturing ingots and ageing annealing shall be verified and utilised in future research. Stabilising effect of Hf on the HCP phase formation during mid temperature ageing shall be verified as well. Influence of the cooling rate on phase composition of HEAs shall be investigated, too. 

Further investigation of the NbTiZr MEA alloy with promising results shall be performed.

Obtained results can be used in evolution of other variants of the investigated system HfNbTaTiZr, for example, for reduction of high temperature oxidation [39] or in research of ageing behaviour of similar RHEAs, especially containing both Ta and Zr.

## 5. Conclusions

Investigations of mechanical properties and microstructure of different MEAs derived from the HfNbTaTiZr alloy system has been performed. It has been shown that ductility is significantly reduced by simultaneous presence of Ta and Zr in the alloy, which leads to reduction of the BCC1 phase content. These elements segregate during solidifying. Because of the miscibility gap in the Ta-Zr equilibrium phase diagram two BCC phases are found after cooling to room temperature. Clear correlation between the BCC1 phase content and the total elongation to failure of the alloy was found. Precipitation of Ta rich precipitates on grain boundaries reduces the elongation to almost zero values. No correlation was found between the tensile mechanical properties and microstructure related parameters in investigated set of alloys. The 5 element HEA alloy exhibits the best combination of strength and elongation. 4 and 3 element MEAs have lower strength to various extent. Some of them are ductile (NbTaTiZr, NbTaTi, NbTiZr) but some of them are brittle (TaTiZr, Nb_1.5_TaTiZr_0.5_, Nb_0.5_TaTiZr_1.5_) depending on microstructure. 

Obtained results on microstructure stability and related mechanical properties can be useful also for long term aging annealing of HEA alloys. The Monte Carlo simulation performed pointed out, that single solid solution of 5 element HfNbTaTiZr alloy is a metastable state at room temperature. Stabilizing effect of Hf on the HCP phase formation during mid temperature aging was proposed.

## Figures and Tables

**Figure 1 entropy-21-00114-f001:**
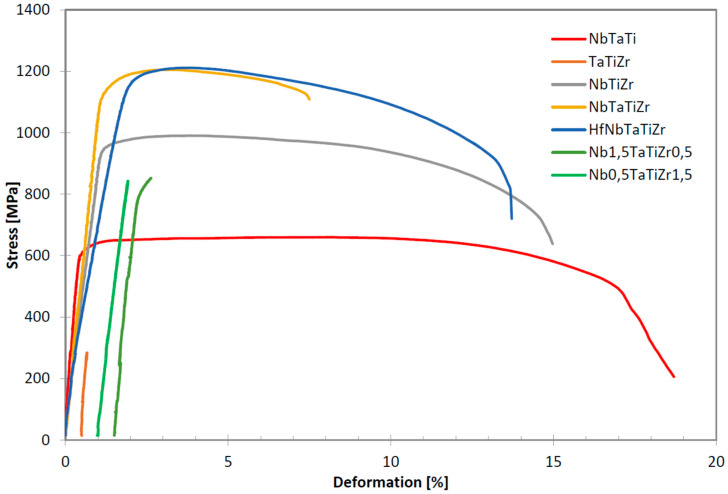
Tensile curves of investigated alloys at room temperature. Curves for brittle alloys TaTiZr, Nb_1.5_TaTiZr_0.5_ and Nb_0.5_TaTiZr_1.5_ were shifted horizontally to make them visible.

**Figure 2 entropy-21-00114-f002:**
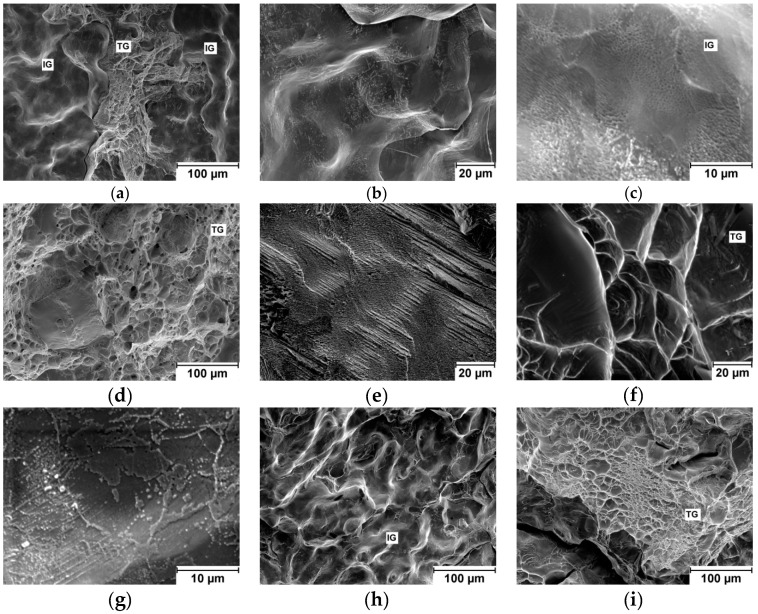
Scanning electron microscopy (SEM) image of investigated fracture surfaces: (**a**), (**b**), (**c**) NbTaTiZr alloy; (**d**) NbTaTi alloy (**e**) TaTiZr alloy; (**f**) NbTiZr alloy; (**g**) Nb_1.5_TaTiZr_0.5_ alloy; (**h**), (**i**) Nb_0.5_TaTiZr_1.5_ alloy; where TG denotes transgranular ductile fracture and IG intergranular ductile fracture.

**Figure 3 entropy-21-00114-f003:**
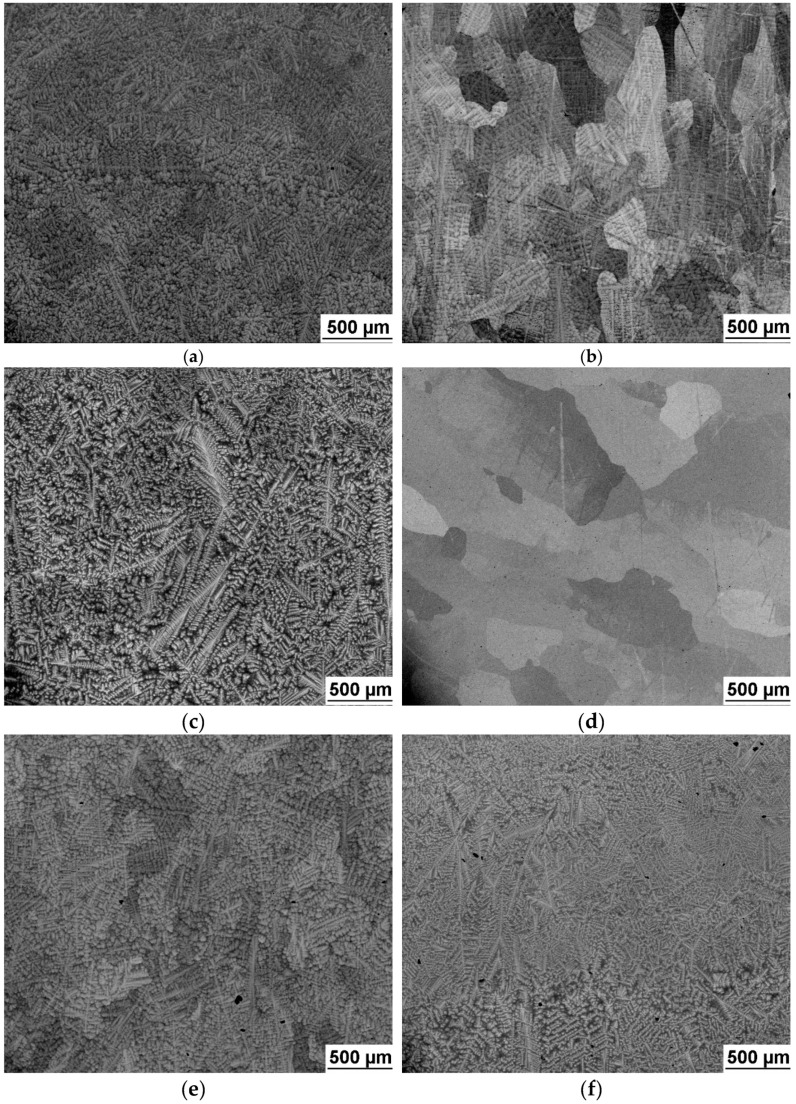
SEM image of investigated alloys microstructure: (**a**) NbTaTiZr alloy; (**b**) NbTaTi alloy; (**c**) TaTiZr alloy; (**d**) NbTiZr alloy; (**e**) Nb_1.5_TaTiZr_0.5_ alloy; (**f**) Nb_0.5_TaTiZr_1.5_ alloy.

**Figure 4 entropy-21-00114-f004:**
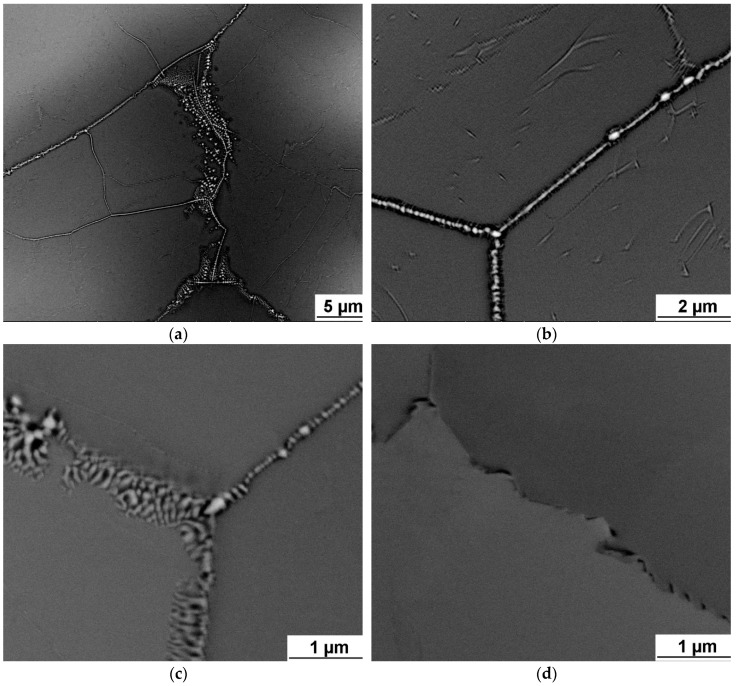
SEM image of investigated alloys microstructure: (**a**), (**b**) TaTiZr alloy; (**c**) Nb_1.5_TaTiZr_0.5_ alloy; (**d**) Nb_0.5_TaTiZr_1.5_ alloy.

**Figure 5 entropy-21-00114-f005:**
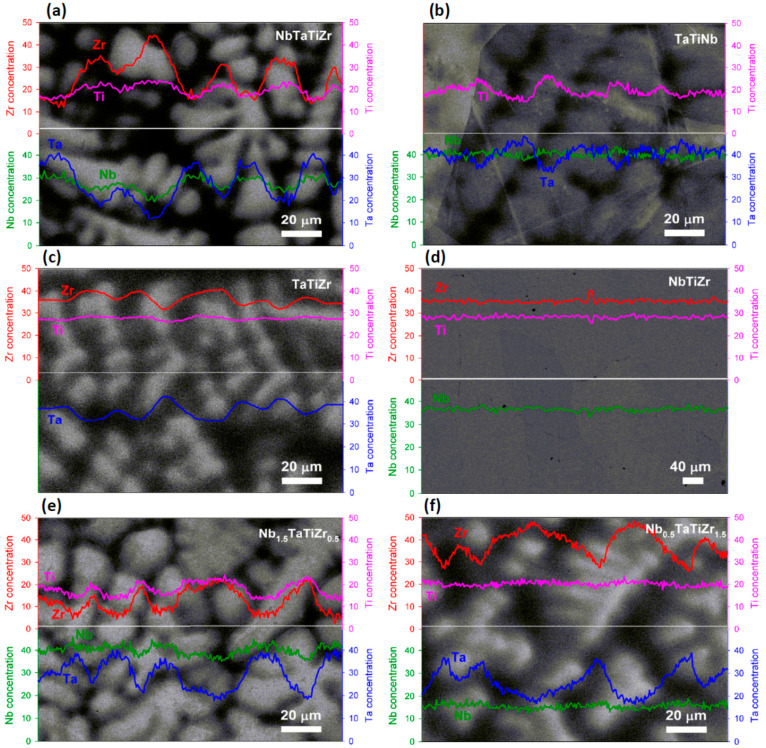
WDS line analysis of: (**a**) NbTaTiZr alloy; (**b**) TaTiNb alloy; (**c**) TaTiZr alloy; (**d**) NbTiZr alloy; (**e**) Nb_1.5_TaTiZr_0.5_ alloy; (**f**) Nb_0.5_TaTiZr_1.5_ alloy.

**Figure 6 entropy-21-00114-f006:**
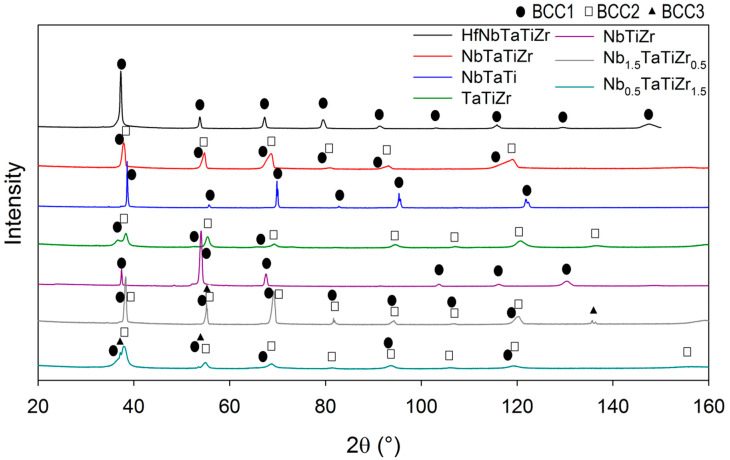
X-ray diffraction (XRD) patterns of investigated alloys. Reflections of the BCC phases are marked by labels.

**Figure 7 entropy-21-00114-f007:**
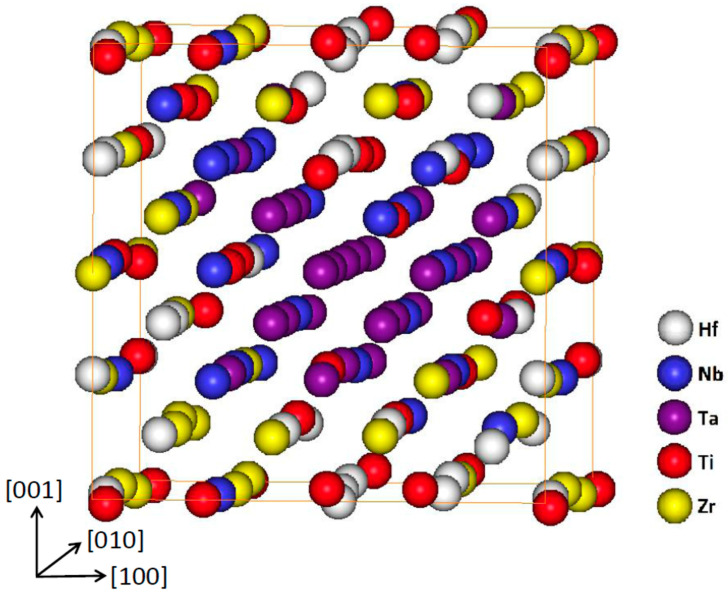
Equilibrium atomic configuration of simulated HfNbTaTiZr alloy.

**Figure 8 entropy-21-00114-f008:**
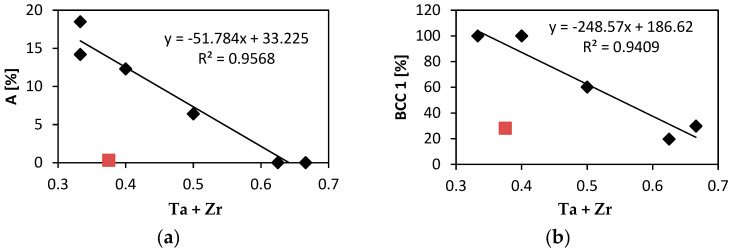
(**a**) Relation of the total elongation to failure A on the sum of Ta and Zr atomic concentration, (**b**) the concentration of the BCC1 phase plotted as a function of the sum of Ta and Zr content. Data for Nb_1.5_TaTiZr_0.5_ alloy are indicated by red symbols and were excluded from linear regression.

**Figure 9 entropy-21-00114-f009:**
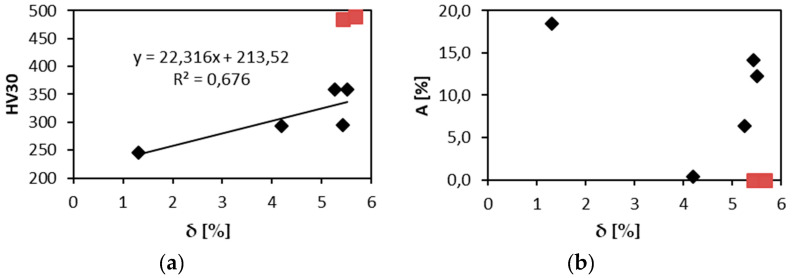
Relation of δ and: (**a**) HV30; (**b**) elongation.

**Figure 10 entropy-21-00114-f010:**
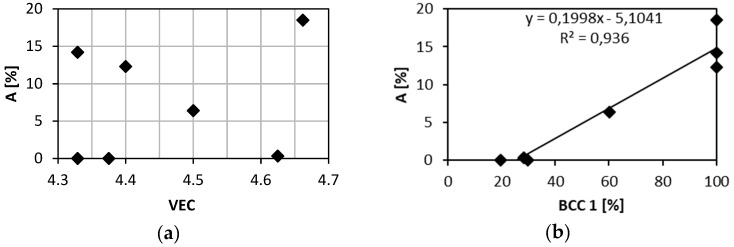
Relation of the elongation to failure and: (**a**) the VEC parameter; (**b**) the content of BCC 1 phase.

**Figure 11 entropy-21-00114-f011:**
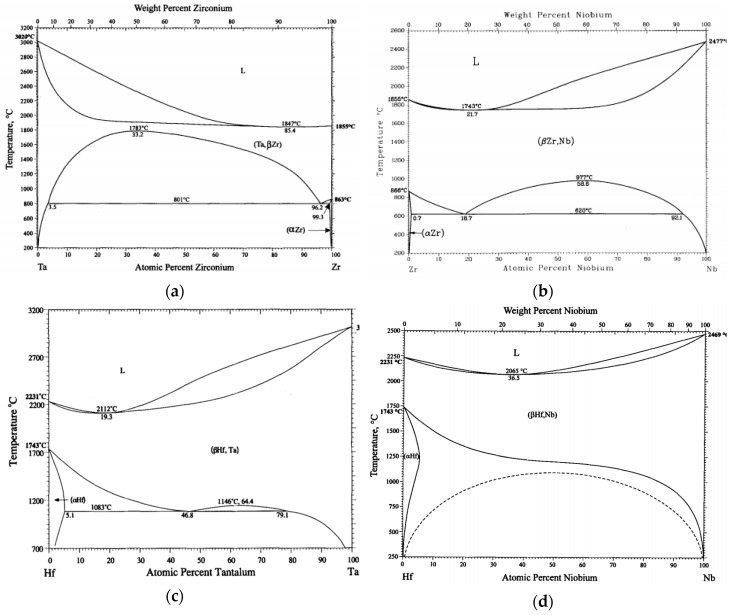
Binary phase diagrams: (**a**) Ta-Zr. Reprinted from [30] with permission of Springer Nature; (**b**) Nb-Zr. Reprinted from [31] with permission of Springer Nature; (**c**) Ta-Hf. Reprinted from [32] with permission of Springer Nature; (**d**) Nb-Hf. Reprinted from [33] with permission of Springer Nature.

**Table 1 entropy-21-00114-t001:** Chemical composition of investigated alloys in at.%, including important properties of individual elements.

	Ti	Ta	Nb	Zr	Hf
molar mass	47.867	180.94	92.9	91.22	91.224
density (g cm^−3^)	4.51	7.14	7.8	6.51	6.5
melting temp. (K)	1941	3269	2741	2125	2495
atomic radius (Å)	1.47	1.43	1.43	1.62	1.6
VEC	4	5	5	4	4
HfNbTaTiZr	0.2	0.2	0.2	0.2	0.2
NbTaTiZr	0.25	0.25	0.25	0.25	0
NbTaTi	0.333	0.333	0.333	0	0
TaTiZr	0.333	0.333	0	0.333	0
NbTiZr	0.333	0	0.333	0.333	0
Nb_1.5_TaTiZr_0.5_	0.25	0.25	0.375	0.125	0
Nb_0.5_TaTiZr_1.5_	0.25	0.25	0.125	0.375	0

**Table 2 entropy-21-00114-t002:** Important high entropy alloy (HEA) related parameters if investigated alloys calculated using data from Table 1.

	VEC	∆H_mix_	δ	T_m_ [K]	Ω	S_mix_	S_mix_	
		[kJ/mol]	[%]			[J/(mol K)]	[R*J/(mol K)]	
HfNbTaTiZr	4.4	2.72	5.51	2514.20	12.37	13.38	1.61	HEA
NbTaTiZr	4.5	2.50	5.26	2519.00	11.61	11.53	1.39	MEA
NbTaTi	4.662	1.33	1.31	2647.68	18.17	9.13	1.10	MEA
TaTiZr	4.329	1.77	5.43	2442.56	12.57	9.13	1.10	MEA
NbTiZr	4.329	2.66	5.43	2266.73	7.78	9.13	1.10	MEA
Nb_1.5_TaTiZr_0.5_	4.625	2.13	4.20	2596.00	13.42	10.98	1.32	MEA
Nb_0.5_TaTiZr_1.5_	4.375	2.38	5.67	2442.00	11.29	10.98	1.32	MEA

**Table 3 entropy-21-00114-t003:** Room temperature mechanical properties of the alloys studied.

Alloy	Rp_0.2_ [MPa]	R_m_ [MPa]	A [%]	E [GPa]	HV30
HfNbTaTiZr	1155	1212	12.3	59	359
NbTaTiZr	1144	1205	6.4	98	358
NbTaTi	620	683	18.5	143	246
TaTiZr	-	284	0	157	485
NbTiZr	956	991	14.2	88	295
Nb_1.5_TaTiZr_0.5_	822	852	0.33	127	294
Nb_0.5_TaTiZr_1.5_	-	843	0	93	489

**Table 4 entropy-21-00114-t004:** Chemical composition in atomic % of bright and dark phase in investigated alloys. Uncertainties of concentrations (one standard deviation) are given in parenthesis.

Alloy	Phase	Ti	Zr	Nb	Ta
Nb_0.5_Ta TiZr_1.5_	Bright	0.19(1)	0.28(1)	0.16(1)	0.37(2)
	Dark	0.22(2)	0.42(3)	0.15(2)	0.21(2)
	nominal	0.25	0.375	0.125	0.25
Nb_1.5_Ta TiZr_0.5_	Bright	0.13(1)	0.07(1)	0.41(3)	0.39(3)
	Dark	0.22(2)	0.22(2)	0.36(2)	0.20(2)
	nominal	0.25	0.125	0.375	0.25
NbTaTiZr	Bright	0.17(1)	0.16(1)	0.32(2)	0.35(22)
	Dark	0.25(2)	0.35(2)	0.23(2)	0.17(1)
	nominal	0.25	0.25	0.25	0.25
NbTaTi	Bright	0.23(2)		0.28(1)	0.49(2)
	Dark	0.42(2)		0.28(1)	0.30(1)
	nominal	0.333		0.333	0.333
TaTiZr	Bright	0.32(2)	0.16(2)		0.52(2)
	Dark	0.40(3)	0.35(3)		0.25(1)
	nominal	0.333	0.333		0.333

**Table 5 entropy-21-00114-t005:** Phase composition of investigated alloys. The lattice parameter (a) and phase content were obtained using Rietveld refinement fitting.

Alloy	BCC1		BCC 2		BCC3	
	a [Å]	[%]	a [Å]	[%]	a [Å]	[%]
HfNbTaTiZr	3.4089(1)	100				
NbTaTiZr	3.3509(8)	60.15	3.380(2)	39.85		
NbTaTi	3.29685(7)	100				
TaTiZr	3.446(1)	29.08	3.3184(2)	70.92		
NbTiZr	3.3969(1)	100				
Nb_1.5_TaTiZr_0.5_	3.3220(5)	28.25	3.334(2)	71.22	3.3273(2)	0.53
Nb_0.5_TaTiZr_1.5_	3.451(5)	19.64	3.3395(3)	77.97	3.4121(4)	2.39

Note: Error of the last digit is shown in the parentheses.

**Table 6 entropy-21-00114-t006:** Mixing enthalpies of element pairs relevant to the investigated alloys [29].

Ti-Ta	Ti-Nb	Ti-Zr	Ti-Hf
1	2	0	0
-	Ta-Nb	Ta-Zr	Ta-Hf
-	0	3	3
-	-	Nb-Zr	Nb-Hf
-	-	4	4
-	-	-	Zr-Hf
-	-	-	0

**Table 7 entropy-21-00114-t007:** Tensile properties of present and reported medium entropy alloys (MEAs) [25].

Alloy	Ta + Zr	HV30	Rp_0.2_ [MPa]	R_m_ [MPa]	A [%]	A calc. [%]	Ref.
HfNbTaTiZr	0.4	359	1155	1212	12.3	12.51	[This work]
NbTaTiZr	0.5	358	1144	1205	6.4	7.33	[This work]
NbTaTi	0.333	246	620	683	18.5	15.98	[This work]
TaTiZr	0.666	485	-	284	0	−1.26	[This work]
NbTiZr	0.333	295	956	991	14.2	15.98	[This work]
Nb_1.5_TaTiZr_0.5_	0.375	294	822	852	0.33	13.81	[This work]
Nb_0.5_TaTiZr_1.5_	0.625	489	-	843	0	0.86	[This work]
NbTaTiZr	0.5	-	* 1190	* 1190	* 0	7.33	[25]
NbTa_0.8_Ti_1.2_Zr	0.45	-	* 1100	* 1110	* 1	9.92	[25]
NbTa_0.6_Ti_1.4_Zr	0.4	-	* 1030	* 1070	* 2.5	12.51	[25]
NbTa_0.4_Ti_1.6_Zr	0.35	-	910	1040	18	15.10	[25]
NbTa_0.2_Ti_1.8_Zr	0.3	-	790	920	22	17.69	[25]

* Values estimated from the figure [25]

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
