# Peer review of "Microstructure and Room Temperature Mechanical Properties of Different 3 and 4 Element Medium Entropy Alloys from HfNbTaTiZr System"

_entropy, 2019, doi:10.3390/e21020114_

Round 1
Reviewer 1 Report
The article reports on fabrication, microstructure and mechanical properties of 3 and 4 element medium 4 entropy alloys from HfNbTaTiZr system. Prior to a publication, the manuscript should be completed and modified according to the following comments.
Comments and recommendations:
1. The English needs a revision:
· Please, change the word “documented” to “observed” or “showed” etc. through the manuscript.
· Line 59. “element table” should be “the periodic table of elements”
2. Abstract. The abstract is too big. Please, reduce the abstract summarizing the most important outcomes.
3. Introduction
a) Page 2, line 45. Avoid this statement “Basic concept is simple”!
b) Page 2, line 45-46. The basic concept of HEAs is not only the obtaining of a stable solid solution. Please, re-formulate to be clear! Refer to the article [3].
c) Page 2, line 47-48. Do you mean the stability of a solid-solution? Or which microstructure? Moreover, solid-solution stability is governed not only by configuration entropy. Refer to [10.1016/j.actamat.2018.01.028] and [/10.1016/j.actamat.2013.01.042]. Please, re-formulate to be clear!
d) Page 2, line 49-53. Please, re-formulate to be clear
e) Page 2, line 54-57. Ceramic HEAs have to be included in this list.
f) Page 2, line 58-59. The melting point of titanium is not the lowest among the elements of IV, V and VI groups. Clarify!
g) Page 2, line 59-61. How the lowest melting point of titanium among the elements can prevail good mutual miscibility? Clarify!
h) Page 2, line 62-63: “some of these alloys can be attractive for bioimplant related materials research”. There are many articles in the literature reporting biocompatible HEAs. Add this information and corresponding references.
i) Also, the metals from IV, V and VI groups of the periodic table were recently used for the synthesis of hard ceramic composites [10.1016/j.matlet.2018.07.048]. This information worth to be added to the introduction.
j) Line 74-76. About what kind of “desired” microstructure and mechanical properties are you writing? Clarify!
k) Page 2, line 77. Re-write the first sentence. Eliminate the word “simple”.
l) Add information on the NbTiZr, TaTiZr and NbTaTi alloys reported in the literature.
4. Results.
· Page 4, line 127-128. Re-write. Grammar mistakes.
· Page 5, line 150-159. Line 147 should be added. Please, refer the text to the corresponding figure. Mark transgranular and intergranular ductile fracture on the figures with arrows. Provide well-visible scale bars for figure 2.
· Line 162-163. How did you measure the grain size by means of SEM? Clarify!
· Line 166-167. There is no evidence that the precipitates visible on Fig. 4 are Ta-rich! SEM images give information on surface morphology only! Explain! Moreover, there are no well-visible precipitates on figure 4c!
· Figure 4. There is no image of the NbTiZr alloy!
· Line 173-174. Fig 4d is Nb0,5TaTiZr1,5 alloy. Clarify!
· Figure 5. Please, increase the font size of the colour captions on the images and put them over corresponding lines!
· Table 4. Please, add errors of the measurements.
· Table 5. Please, add the calculated grain sizes for each phase using XRD results.
· Line 202-205 should be transferred to the Materials and Methods section.
· Figure 7. Please, indicate directions on the image.
5. Discussion.
a) Line 214-217. Are these statements valid for all alloys studied? Clarify!
b) Line 247-248. The sentence is not clear. Re-formulate.
c) Please, add a table or a figure where compare your results (hardness, strength and elongation to failure) with those reported by other authors for these alloys.
The article contains very valuable experimental outputs on mechanical properties and microstructure of different MEAs derived from the HfNbTaTiZr alloy system, but the major revision is necessary before publication.
Author Response
The authors would like to thank both referees for their valuable comments. The paper has been
subjected to major revision in the view of the referee’s comments. All comments of both referees
were carefully addressed and the manuscript has been corrected or amended appropriately.
Modified parts of the manuscript are highlighted by blue color for easy orientation. Detailed author’s
response to each referee’s comment is given in the following file.

Reviewer 2 Report
This paper studied the microstructures and room-temperature tensile properties of the medium entropy alloys. It is very interesting. However, I would suggest to publish it considering several minor issues:
1. SEM and WDS are microscope and spectroscope, not microscopy and spectroscopy.
2. Fig.1 shows only five tensile curves, the curves of TaTiZr and Nb0.5TaTiZr1.5 are not shown in this figure.
3. In Fig.6, all of the peaks should be marked to show the phases.
4. Table 5 lists the lattice constants of the phases and their percentages of content. How to measure the percentage of each phase should be described in the manuscript.
Author Response

(The authors gave the same response as above.)

Round 2
Reviewer 1 Report
The revised manuscript entropy-415677 was modified and competed according to my comments and recommendations. However, they still have to address the following points:
1) The English (grammar, spelling…) needs to be further improved throughout the manuscript with the help of a native English speaker.
2) The affiliation section contains repetitions. Clarify!